# HPRT1 Promotes Chemoresistance in Oral Squamous Cell Carcinoma via Activating MMP1/PI3K/Akt Signaling Pathway

**DOI:** 10.3390/cancers14040855

**Published:** 2022-02-09

**Authors:** Tong Wu, Zan Jiao, Yixuan Li, Xuan Su, Fan Yao, Jin Peng, Weichao Chen, Ankui Yang

**Affiliations:** 1Department of Head and Neck Surgery, Sun Yat-sen University Cancer Center, Guangzhou 510060, China; wutong@sysucc.org.cn (T.W.); jiaozan@sysucc.org.cn (Z.J.); liyx@sysucc.org.cn (Y.L.); suxuan@sysucc.org.cn (X.S.); yaofan@sysucc.org.cn (F.Y.); pengjin1@sysucc.org.cn (J.P.); chenwch@sysucc.org.cn (W.C.); 2State Key Laboratory of Oncology in South China, Collaborative Innovation Center for Cancer Medicine, Guangzhou 510060, China

**Keywords:** hypoxanthine phosphoribosyl transferase 1, housekeeping gene, chemoresistance, matrix metalloproteinase 1, PI3K/AKT signaling pathway

## Abstract

**Simple Summary:**

Hypoxanthine phosphoribosyl transferase 1 (HPRT1) is traditionally believed to be a housekeeping gene, however, we found that highly expressed HPRT1 was associated with a poor prognosis and could promote resistance to cisplatin (CDDP) in OSCC cells in both in vitro and in vivo. Hence, HPRT1 can no longer be simply believed to be a housekeeping gene. HPRT1 over- expression indicates a worse prognosis and can improve CDDP resistance for patients with OSCC by promoting the MMP1/PI3K/Akt axis, and it may be a potential prognostic biomarker and thera- peutic target in OSCC.

**Abstract:**

Hypoxanthine phosphoribosyl transferase 1 (HPRT1) is traditionally believed to be a housekeeping gene. However, recent reports have indicated that HPRT1 overexpression is associated with a poor prognosis in various types of cancers. Using The Cancer Genome Atlas (TCGA), HPRT1 was found to be highly expressed in various cancer types, especially in head and neck squamous cell carcinoma (HNSCC). Therefore, we measured HPRT1 expression in human cancer tissues and adjacent non-carcinoma tissues (ANT) and explored the relationship between HPRT1 expression and clinical pathological factors and prognosis in patients with oral squamous cell carcinoma (OSCC), a common type of HNSCC. We built OSCC cells with stable knockdown and overexpression of HPRT1 to observe its influence on chemoresistance and malignancy in vitro and vivo. We found that highly expressed HPRT1 was associated with a poor prognosis and could promote resistance to cisplatin (CDDP) in OSCC cells in both in vitro and in vivo. An RNA sequence assay was carried out to explore the mechanism of function of HPRT1, we found that HPRT1 could positively regulate the expression of MMP1 and the activation of the PI3K/AKT pathway, to regulate the resistance to CDDP of OSCC. In conclusion, HPRT1 can no longer be simply believed to be a housekeeping gene. HPRT1 overexpression indicates a worse prognosis and can improve CDDP resistance for patients with OSCC by promoting the MMP1/PI3K/Akt axis. HPRT1 may be a potential prognostic biomarker and therapeutic target in OSCC.

## 1. Introduction

Oral squamous cell carcinoma (OSCC) is a common type of head and neck squamous cell carcinoma (HNSCC), which ranked as the sixth most common malignant tumor around the world with a poor prognosis, a high incidence rate, and a mortality rate [1]. The high rate of recurrence and distant metastasis results in cancer-related deaths and a five-year survival rate of OSCC of less than 50%. Despite the evolving development of treatment technology, cisplatin (CDDP) remains the first-line treatment regimen in patients with OSCC with recurrence and progression [2,3]. However, CDDP resistance is a huge obstacle to the efficacy of OSCC treatment, and can lead to early recurrence and metastasis. Unfortunately, biomarkers of resistance to CDDP are rare. Therefore, it is significant to explore the molecular mechanism of CDDP resistance, to identify effective predictors.

Hypoxanthine phosphoribosyl transferase 1 (HPRT1) encodes an enzyme, which mainly participates in the cell cycle through the regulation of purine and inosine production in the salvage pathway [4,5]. HPRT1 is an important protein that provides the necessary components for cell growth and is highly expressed in most tissues. HPRT1 has traditionally been used as a housekeeping gene; however, many recent studies have found a potential relationship with various types of cancer [6,7,8]. HPRT1 overexpression was significantly correlated with the poor prognosis of patients with HNSCC [7]. HPRT1 was also found to be associated with drug resistance in cancer cells and has been proposed as a therapeutic target of chemotherapeutic drugs [9]. The antitumor effects of CDDP occur mainly through DNA damage; while the main role of HPRT1 is that of providing materials for DNA synthesis through the rescue pathway. Therefore, we studied the relationship between HPRT1 and the prognosis of OSCC and explored its relationship with response to CDDP treatment.

In this study, we found that overexpressed HPRT1 can not directly affect the proliferation of OSCC cells but we revealed thatit is significantly overexpressed and associated with CDDP chemoresistance in OSCC using both in vivo and in vitro experiments. Furthermore, HPRT1 can positively regulate the expression of MMP1, which belongs to a family of 23 extracellular endopeptidases with enzyme activity against almost all protein components of the extracellular matrix (ECM), which have been implicated in the invasion and metastasis of HNSC [10]. Importantly, the MMP1/PI3K/AKT axis proved to be important for HPRT1-induced chemoresistance in OSCC cells. Our findings reveal an innovative mechanism of chemoresistance of OSCC, and show that HPRT1 may act as a potential prognostic marker and anti-resistance therapeutic target.

## 2. Materials and Methods

### 2.1. Ethics Statement

The Ethics Committee of the Sun Yat-sen University Cancer Center approved the use of tumor specimens used for this research and OSCC patients signed consent forms.

### 2.2. Patients and Tumor Tissue Samples

Specimens in this study were collected from 136 patients who were diagnosed with OSCC at Sun Yat-sen University Cancer Center. All patients underwent biopsy before surgery and the Department of Pathology of Sun Yat-sen University Cancer Center confirmed the diagnosis of OSCC following the histological criteria of the World Health Organization (WHO). Six pairs of fresh OSCC tissues were obtained from diagnosed patients during surgical intervention, while the respective adjacent normal tissues were sampled 5 cm away from the tumor lesion.

### 2.3. Cell Culture

Human OSCC cell lines (SCC25, Cal27, UM1, HSC3, SCC15, SAS, Cal33) and the normal cell line NOK were purchased from iCell Bioscience Inc. (Shanghai, China). All cell lines were cultured in Dulbecco’s modified Eagle’s Medium (DMEM; Invitrogen, Carlsbad, CA, USA) supplemented with 10% fatal bovine serum (FBS; Sigma-Aldrich, St. Louis, MO, USA). Short tandem repeat (STR) was reported in all cell lines. All cell lines were kept in a humidified atmosphere of 5% CO2 at 37 °C.

### 2.4. RNA Isolation and Quantitative Real-Time Reverse Transcription PCR

RNA was extracted with Trizol. Quantitative reverse transcription polymerase chain reaction (qRT-PCR) was performed using CFX96 real-time system C1000 thermocycler (Bio- Rad laboratory, Singapore). The expression of GAPDH, encoding glyceraldehyde-3-phosphate dehydrogenase, was used as the internal housekeeping gene. The primers used in this study are shown in Appendix A.

### 2.5. Constructs, Transfection, and Retroviral Infection

When the cells reached a confluence of 50–70% of the 6- well culture dish, four siRNAs (siRNA-NC, siRNA#1, siRNA#2, and siRNA#3) (Ribobio Inc., Guangzhou, China) were transfected into the cells in different wells using the transfection reagent RNAiMax (Thermo Fisher Scientific, Massachusetts, Waltham, MA, USA). After 8 h, the medium was replaced with DMEM medium with 10% FBS. Cells were collected 48 h later for the subsequent assays. ShRNA vector and HPRT1 overexpression plasmid of HPRT1(GeneCopoeia Inc., Baltimore, MD, USA) were transfected into designated cells with transfection reagent Lipo3000 (Thermo Fisher Scientific, Massachusetts, Waltham, MA, USA) respectively. Transfected cells were cultured with puromycin (500 μg/mL) for 7 days and stable cell lines with HPRT1 overexpression or suppression were established. The sequences of siRNAs and shRNAs used in this study are shown in Appendix A.

### 2.6. Total RNA Isolation, RNA-Seq Library Preparation, and Sequencing

Three treatment conditions (si-NC, siRNA#1, and siRNA#2) in the Cal27 the cell line were used for RNA-seq analyses. Total RNA was extracted according to the manufacturer’s instructions (EZ-10 DNA away RNA Miniprep kit, BS88136). RNA was eluted with RNAse-free water and was assayed for quality by electrophoresis (Agilent RNA ScreenTape). Only samples with RIN (RNA integrity number) values of >9 were included in the subsequent analysis. RNA-seq libraries were prepared and sequenced at Majorbio Corporation (Shanghai, China).

### 2.7. Western Blotting

The total protein was harvested in sample buffer. The protein concentration was quantified using the bicinchoninic acid (BCA) method test kit according to the manufacturer’s instructions. Equal volume of proteins from each sample was electrophoretically separated on SDS/Polyacrylamide gels and was then transferred to polyvinylidene difluoride (PVDF) membranes (Roche). The PVDF membrane was incubated with primary antibody at 4 °C overnight. The primary antibodies used in this study are shown in Appendix A. GAPDH was used as an internal reference protein.

### 2.8. Immunohistochemistry

A total of 136 paraffin embedded OSCC tissue sections were used for immunohistochemistry (IHC) with an anti-HPRT1 antibody. The staining intensity was divided into four grades: no staining (0), weak staining (1), medium staining (2), and strong staining (3). The percentage of positive cells was also stratified into four grades: 0% (0), 1–5% (1), 6–29% (2), 30–59% (3), and more than 60% (4). Finally, HPRT1 staining score was calculated as = positive cell score + staining intensity score. For statistical analysis, each sample was grouped as positive (score > 3) or negative (score 0–3).

### 2.9. Annexin V-Fluorescein Isothiocyanate (FITC)/Propidium Iodide (PI)-Staining Assay

The cell apoptotic rate was analyzed by flow cytometry according to the standard manufacturer’s protocol (MACS Miltenyi Biotec, Bergisch Gladbach, Germany). Cells were trypsinized, washed in ice-cold PBS, and centrifuged at 1000× *g* for 5 min. The pellet was resuspended in binding buffer at a density of 1.0 × 10^6^ cells/mL and incubated with Annexin V-FITC (BD Biosciences, San Jose, CA, USA) and PI for 25 min at 4 °C in the dark. After staining, cells were analyzed by flow cytometry (Cytomics FC500; Beckman Coulter, Miami, FL, USA). Apoptotic cells were calculated as the proportion of Annexin V-positive cells

### 2.10. Cell Viability Assay

After 24 h of transfection, OSCC cells were seeded into 96-well plates and cultured for 7 days. Next, 10 μL cell counting kit-8 (CCK-8) solution (Dojindo, Tokyo, Japan) was added to each well every day. Cell viability was assessed by measuring absorbance at 450 nm.

### 2.11. Colony Formation Assay

Transfected cells (600 cells per well) were inoculated into 6-well plates. They were cultured in DMEM supplemented with 10% FBS and CDDP (2ug/mL). After 2 weeks of incubation, the cells were fixed in paraformaldehyde and stained with 0.1% crystal violet. The colonies were counted using Quantity software (BioRad, Hercules, CA, USA). All experiments were repeated three times and the average value was calculated.

### 2.12. TUNEL Assay

The instructions of the TUNEL kit (Elabscience Biotechnology Co., Ltd., Wuhan, China) was followed for all experiments. Paraffin sections of mouse tissues were first dewaxed with a gradient xylene solutions. The working solution was also prepared according to the instructions, and 50 μL TdT Equilibration Buffer was added. Finally, the sections were stained with DAPI before observation under fluorescence microscope.

### 2.13. Bioinformatics Analysis

Transcripts with log2 fold change cutoff of 1.5 and *p*-value ≤ 0.05 were considered as significantly differentially expressed. And KEGG pathway analysis were carried out by KOBAS (http://kobas.cbi.pku.edu.cn/home.do) [11]. The analysis was accessed and graphed on 2 June 2021. The clinical data and overall survival (OS) time for TCGA head and neck cancer patients were collected from the UCSC Xena website (UCSC Xena. Available online: http://xena.ucsc.edu/welcometo-ucsc-xena/). Microarray results with accession numbers GSE42743, which was performed by Holsinger C et. al. from Stanford University School of Medicine to compare differences of gene expression between oral cancer samples and adjacent normal mucosa.

## 3. Statistical Analysis

Statistical analysis was performed using SPSS software version 26.0 (IBM Corp., Armonk, NY, USA). Statistical tests included the chi-squared test, Spearman’s rank correlation analysis, and the student’s *t*-test (two tailed). The survival curve was drawn by Kaplan-Meier analysis. Multivariate Cox regression analysis was used to analyze the correlation between HPRT1 expression and various clinicopathological parameters. *p* < 0.05 was considered statistically significant.

## 4. Results

### 4.1. Overexpressed HPRT1 Predicted a Poor Prognosis in OSCC Patients in TCGA

First, we applied the TIMER2 approach to analyze the expression status of HPRT1 in various types of cancer of TCGA. The expression of HPRT1 in different cancers was significantly higher than that of the corresponding control tissues (Figure 1A). Next, we found that HPRT1 mRNA levels were markedly upregulated in tumor tissues compared to adjacent normal tissues (Figure 1B). Importantly, Kaplan-Meier survival curves showed that patients with high expression of HPRT1 had worse overall survival (OS) and disease-free survival (DFS) compared to those patients with low expression of HPRT1 (Figure 1C), indicating that HPRT1 is a potential prognostic index in HNSCC. Similarly, the GSE42743 results and Kaplan-Meier survival analysis also demonstrated that HPRT1 expression was upregulated and was associated with a poor prognosis (Figure 1D,E) in OSCC. All the above results showed that overexpressed HPRT1 predicted a poor prognosis in patients with OSCC.

### 4.2. HPRT1 Is a Potential Prognostic Indicator in OSCC

To investigate HPRT1 expression in oral tissues, we collected 6 pairs of fresh primary OSCC tissues and adjacent non-neoplastic tissues from patients at our cancer center. The results showed that HPRT1 was highly expressed in all OSCC tissues compared to adjacent paired non-neoplastic tissues both in mRNA (Appendix A) and protein levels (Figure 2A and Appendix A). Seven OSCC cell lines showed significantly upregulated endogenous expression of HPRT1 mRNA and protein compared to the normal oral cell line NOK examined by qRT-PCR (Appendix A)and western blotting (Figure 2B and Appendix A). These results suggested that up-regulated expression of the HPRT1 protein may be a tumor-specific event. Next, IHC studies were applied in 136 OSCC tissues to explore the correlation between HPRT1 expression and various clinicopathological parameters, and we found that high HPRT1 expression was associated with poor clinical outcomes (Appendix A). Univariate and multifactorial analyses identified high HPRT1 expression as an independent prognostic factor that affected both DFS and OS (Appendix A and Appendix A). According to its staining intensity, the specimens were defined as HPRT1-high (68 cases) and HPRT1-low (68 cases). In particular, statistical analysis showed that HPRT1 increased significantly with the development of OSCC (Figure 2C). Furthermore, correlation analysis showed that high HPRT1 expression was associated with a high clinical stage, recurrence status, and poor survival status (Figure 2D). Importantly, the Kaplan-Meier survival curve and the log-rank test showed that 5-year DFS and OS in patients with high expression of HPRT1 were poor, which indicated that HPRT1 is a potential prognostic indicator (Figure 2E). These results suggested that HPRT1 was associated with the clinical stage of OSCC and predicted a poor prognosis.

### 4.3. Knockdown of HPRT1 Reduced Cisplatin Resistance of Oral Cancer Cells In Vitro

To verify the effect of HPRT1 on oral carcinoma cells, we used three specific sh-RNAs targeting HPRT1, and both could efficiently reduce the expression of HPRT1 in Cal27 and SCC25 cells by qRT-PCR and western blotting (Figure 3A and Appendix A). Colony survival and cell viability assays showed that HPRT1 silencing dramatically interfered with cell viability and proliferation under CDDP treatment, indicating that HPRT1 knockdown increased the therapeutic efficacy of CDDP (Figure 3B,C). Furthermore, the apoptosis assay further demonstrated that HPRT1 silencing increased CDDP-induced apoptosis, suggesting that HPRT1 participates significantly in OSCC cell chemoresistance (Figure 3D). All the above results suggested that HPRT1 was an important positive regulator of chemoresistance in OSCC and silencing of HPRT1 may re-sensitize chemotherapy-resistant OSCC patients.

### 4.4. HPRT1 Positively Regulates the Expression of MMP1 in Oral Cancer Cells

Three siRNAs (Si-NC, Si-2, and Si-3) treated Cal27 cell groups were subjected to RNA-Seq analysis. The genes that were differentially regulated by both siRNAs were chosen. The top five genes with greatest fold change (HFM1, CDRT1, TECRP1, FSCN2, MMP1) were chosen (Figure 4A and Appendix A). MMP1 was identified as a potential HPRT1 target gene by western blotting analysis (Figure 4B). At the same time, the correlation analysis showed that there was a strong correlation between HPRT1 and MMP1 mRNA expression levels in the GSE42743 dataset (Figure 4C). To confirm this relationship, we detected the expression of HPRT1 and MMP1 at gene and protein levels in the Cal27 and SCC25 cell lines. The results showed that MMP1 was down regulated at the mRNA level in Cal27 cells transfected with sh-HPRT#1. Meanwhile, HPRT1 overexpression (HPRT-OE) led to the simultaneous increase in the levels of MMP1 in SCC25 cells. A similar trend was observed at the protein level (Figure 4D and Appendix A). The above results suggested that MMP1 is a potential target gene of HPRT1. After the suppression of MMP1 in SCC25 cells by HPRT1 overexpression, drug resistance to CDDP was increased. Colony survival and cell viability assays and apoptosis assays provided evidence that HPRT1 promoted OSCC drug resistance by regulating MMP1 expression (Figure 4E–G).

### 4.5. HPRT1 Promote Chemoresistance In Vivo

We explored the role of HPRT1 in vivo using a nude mouse model. Stable cells with different levels of HPRT1 expression (SCC25-OE, SCC25-Sh#1) and control cells (SCC25-Vector, SCC25-Control) were inoculated subcutaneously in nude mice. CDDP was injected in the second week; The tumor growth volume of each group was measured (Figure 5A). The tumors of nude mice in the overexpression group increased most significantly and rapidly in size, followed by the control group and the sh#1 group. Tumor volumes were measured weekly and the tumor was weighed after 6 weeks (Figure 5B). IHC also determined that the expression of HPRT1 and MMP1 were consistent in the tumor tissues of nude mice. Further, the expression of HPRT1 correlated with γ-H2AX expression, and γ-H2AX expression was highest in the HPRT1-Sh#1 group and lowest with HPRT1-OE group (Figure 5C). These findings indicated that HPRT1 could enhance CDDP drug resistance by positively regulating the expression of MMP1. Terminal deoxynucleotidyl transferase-mediated dUTP Nick-End labeling (TUNEL) experiments showed that with regard to HPRT1 expression, the proportion of apoptotic cells in the HPRT1-OE group were the highest, while the HPRT1-Sh#1 group had the smallest number of apoptotic cells (Figure 5D). Altogether these findings indicated that HPRT1 plays a key role in CDDP resistance by regulating MMP1.

### 4.6. HPRT1 Promoted CDDP Chemoresistance in OSCC by Activating the PI3K/AKT Pathway

KEGG pathway analysis showed that genes related to HPRT1 and MMP1 were enriched in the PI3K/Akt pathway (Figure 6A). The PI3K/Akt pathway has been shown to be related to chemoresistance. Therefore, we hypothesized that HPRT1 can promote OSCC chemoresistance by inducing the enhanced activation of the MMP1/PI3K/AKT pathway. The results of western blotting also confirmed that phosphorylated Akt/MMP1 was up-regulated and associated with OSCC survival and c-myc expression (Figure 6B and Appendix A). We treated Cal27 of shRNAs and SCC25 cells overexpressing HPRT1 with LY294002, an inhibitor of the PI3K/Akt pathway. Clone formation and cell viability experiments confirmed that the CDDP chemoresistance of SCC25-OE-HPRT1 cells with LY294002 was significantly weakened (Figure 6C,D). These results suggest that HPRT1 promoted the chemoresistance of OSCC, at least in part, via activation of the PI3K/AKT pathway.

## 5. Clinical Value of HPRT1 in Other Cancer Types

We applied the GEPIA2 approach to analyze the expression status of HPRT1 across various cancer types of TCGA. The expression level of HPRT1 in most tumor tissues was significantly higher, especially in breast invasive carcinoma (BRCA), cervical squamous cell carcinoma and endocervical adenocarcinoma (CESC), colon adenocarcinoma (COAD), esophageal carcinoma (ESCA), lymphoid neoplasm diffuse large B-cell lymphoma (DLBC), lung adenocarcinoma (LUAD), skin cutaneous melanoma (SKCM), stomach adenocarcinoma (STAD), lung squamous cell carcinoma (LUSC), pancreatic adenocarcinoma (PAAD), thymoma (THYM), and uterine corpus endometrial carcinoma (UCEC) compared to the corresponding control tissues (Figure 7A). We found that highly expressed HPRT1 was also associated with poor prognosis and the OS of kidney renal papillary cell carcinoma (KIRP) (*p* = 0.00026), brain lower grade glioma (LGG) (*p* = 0.031), liver hepatocellular carcinoma (LIHC) (*p* = 0.017), uterine carcinosarcoma (UCS) (*p* = 0.0091), lung adenocarcinoma (LUAD) (*p* = 0.023), lung squamous cell carcinoma (LUSC) (*p* = 0.023), mesothelioma (MESO) (*p* = 3.3 × 10^−6^) within the TCGA project (Figure 7B). Our analysis also showed that high HPRT1 expression also associated with poor prognosis in the DFS analysis using data from TCGA cases of (uveal melanoma) UVM (*p* = 0.018), adrenocortical carcinoma (ACC) (*p* = 0.017), KIRP (*p* = 0.086), sarcoma (SARC) (*p* = 0.018), UCEC (*p* = 0.014), and LUAD (*p* = 0.0086) (Figure 7C). The above indicated that HPRT1 expression is differentially associated with the prognosis of cases with different tumors.

## 6. Discussion

CDDP-containing chemotherapy regimens are the first-line treatment for patients with advanced and recurrent metastatic OSCC [2,3,12]. However, resistance to CDDP affects the therapeutic outcomes of a considerable number of patients. Therefore, it is very important to study the molecular mechanisms responsible for CDDP resistance in OSCC and to develop new biomarkers of CDDP resistance. The present study suggested that up-regulation of HPRT1 in OSCC was related to resistance to CDDP, indicating that HPRT1 could be a potential predictor of the therapeutic efficacy. Silencing HRPT1 expression could sensitize resistant OSCC to CDDP treatment, further suggesting HRPT1 as a potential therapeutic target in CDDP-resistance. Importantly, HRPT1 was a marker of clinical prognosis, because the upregulation of HRPT1 was associated with poorer OS and DFS in patients with OSCC.

HPRT1 has traditionally been considered a housekeeping gene and is widely used as an endogenous control in gene expression studies. However, increasing evidence has indicated the differential expression of HPRT1 and its urgent role in cancer cells due to the increased demand for nucleotide synthesis and consequently HPRT1 during the cell cycle [12]. In our study, we found that mRNA levels of HPRT1 were significantly different in oral cancer tissues and corresponding adjacent tissues, and was much higher in cancer tissues. The same results were also found in various OSCC cell lines and normal oral epithelial cells. Therefore, we cannot simply regard HPRT1 as an endogenous control gene, at least not in HNSCC.

Matrix metalloproteinases (MMPs) are a family of zinc-dependent endopeptidases and are overexpressed in various malignant tumors. MMP1 promotes proliferation and drug resistance in many cancers, such as lung cancer [13], breast cancer [14], and head and neck cancer [15]. Therefore, targeting MMP1 is considered an important step in reversing drug resistance. Our study indicated that HPRT1 could interfere with CDDP activity by regulating MMP1. However, the specific mechanism by which HPRT1 regulated MMP1 is unknown, which will be the focus of our future research. MMP1 was reported to be associated with CDDP resistance, which is a major obstacle in patients with recurrent and advanced cancer [16]. PI3K/AKT pathway is a classic cancer pathway, which can cause cell proliferation, metastasis, and drug resistance and resistance to radiotherapy [17,18]. In addition, cancer cell sensitivity to CDDP treatment can be restored by inhibiting the PI3K/AKT pathway [19,20,21]. In our study, we demonstrated that MMP1 is a downstream gene of HPRT1 and can affect the CDDP resistance of OSCC by activating the PI3K/AKT pathway in both in vivo and in vitro experiments. Therefore, in the treatment of OSCC, blocking the HPRT1/MMP1/PI3K/AKT axis might be an important approach to reverse resistance to CDDP in oral cancer.

Our study found that HPRT1 is highly expressed in many common cancers and corresponds to a poor clinical prognosis, and its malignancy and expression of resistance to CDDP in oral cancer has been identified, but there is very little research on HPRT1 in other tumor areas. Therefore, further research is warranted focusing on the role of HPRT1 as a potential oncogene, which may have important implications for cancer treatment.

## 7. Conclusions

HPRT1 should no longer be considered as a housekeeping gene. HPRT1 is highly expressed in many common cancers and corresponds to a poor clinical prognosis. Our findings showed that over-expression of HPRT1 can enhance CDDP resistance and malignancy in patients with OSCC by promoting the MMP1/PI3K/AKT signaling pathway. Further, HPRT1 may represent a potential prognostic biomarker and therapeutic target in OSCC. In the treatment of OSCC, blocking the HPRT1/MMP1/PI3K/AKT axis could be an important approach to reverse resistance to CDDP.

## Figures and Tables

**Figure 1 cancers-14-00855-f001:**
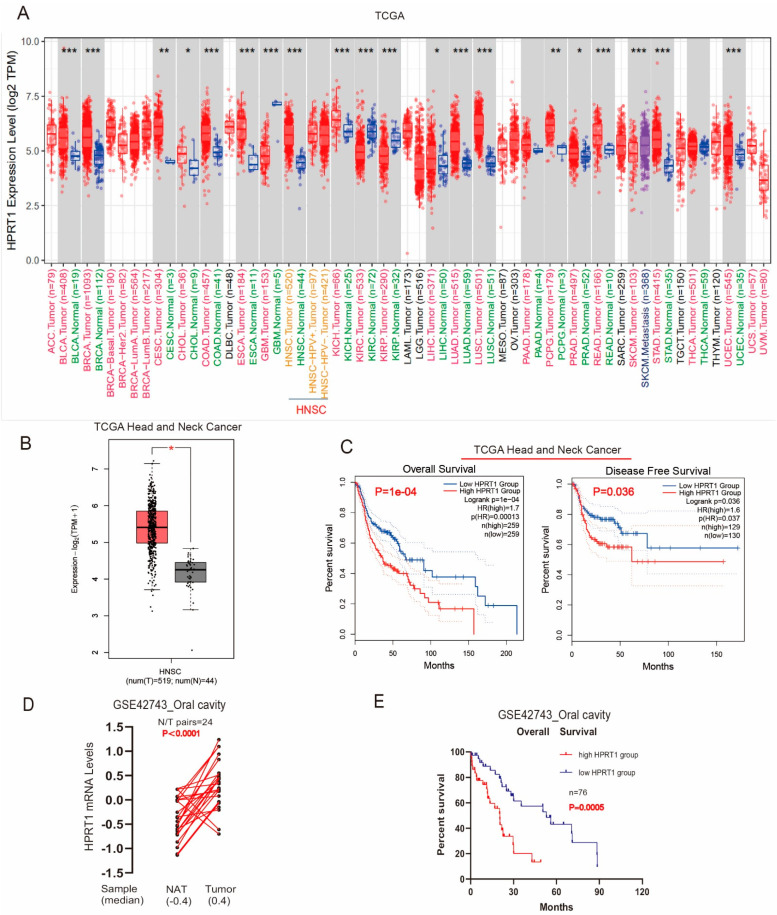
Overexpressed HPRT1 predicted poor prognosis in OSCC patients in TCGA. (**A**) HPRT1 is highly expressed in most cancer species. (**B**) Comparison of mRNA levels of HPRT1 in HNSC and adjacent normal tissues. (**C**) OS and DFS curves of different HPRT1 expression levels in HNSC. (**D**) mRNA levels of HPRT1 in 21 pairs of OSCC and adjacent normal tissues from GSE42743. (**E**) OS curve of HPRT1 expression level in 76 OSCC cases in GSE42743. OS, overall survival; DFS, disease-free survival; OSCC, oral squamous cell carcinoma; HNSC, head and neck squamous carcinoma; Values present the mean ± SEM. * *p* < 0.05, ** *p* < 0.01, *** *p* < 0.001.

**Figure 2 cancers-14-00855-f002:**
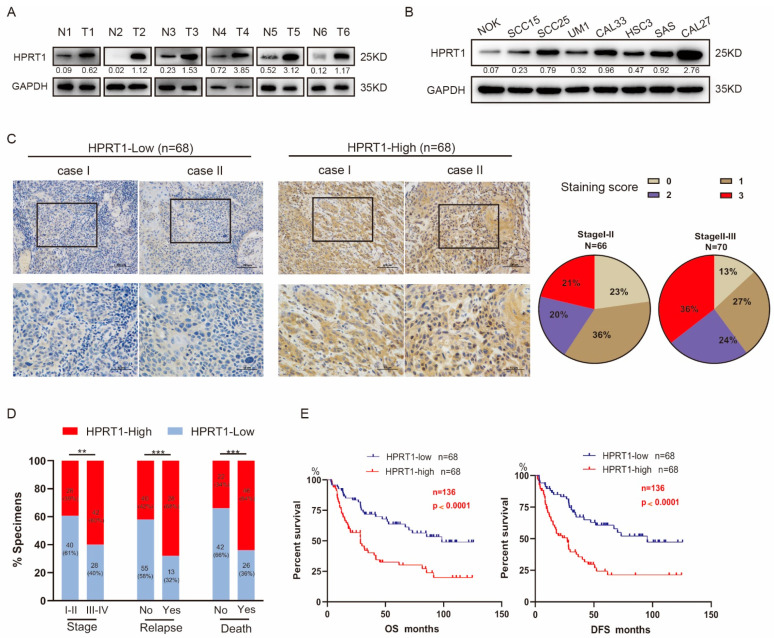
HPRT1 is a potential prognostic indicator in OSCC. (**A**) Western blotting of HPRT1 protein level in OSCC tissues (T) and matched adjacent non-tumor tissues (N) in 6 patients with OSCC. Glyceraldehyde-3-phosphate dehydrogenase (GAPDH) was regarded as a loading control. (**B**) Expression of HPRT1 protein levels in NOK (normal oral mucosal epithelial cell) versus different OSCC cell lines(SCC25, Cal27, UM1, HSC3, SCC15, SAS, Cal33). (**C**) Representative images of HPRT1 IHC staining in OSCC tissues, classified according to staining intensity as HPRT1-High and HPRT1-Low. (left panel); the distribution of intensity score in stage I-II and stage III-IV (right panel). (**D**) Correlation analysis of HPRT1 expression level with tumor stage, relapse status and survival status of patients (χ^2^ test). (**E**) Kaplan-Meier analysis of disease-free survival (DFS) and overall survival (OS) of patients with OSCC stratified based on their HPRT1 expression levels. Values present the mean ± SEM. ** *p* < 0.01, *** *p* < 0.001. We repeat-ed every experiments for 3 times.

**Figure 3 cancers-14-00855-f003:**
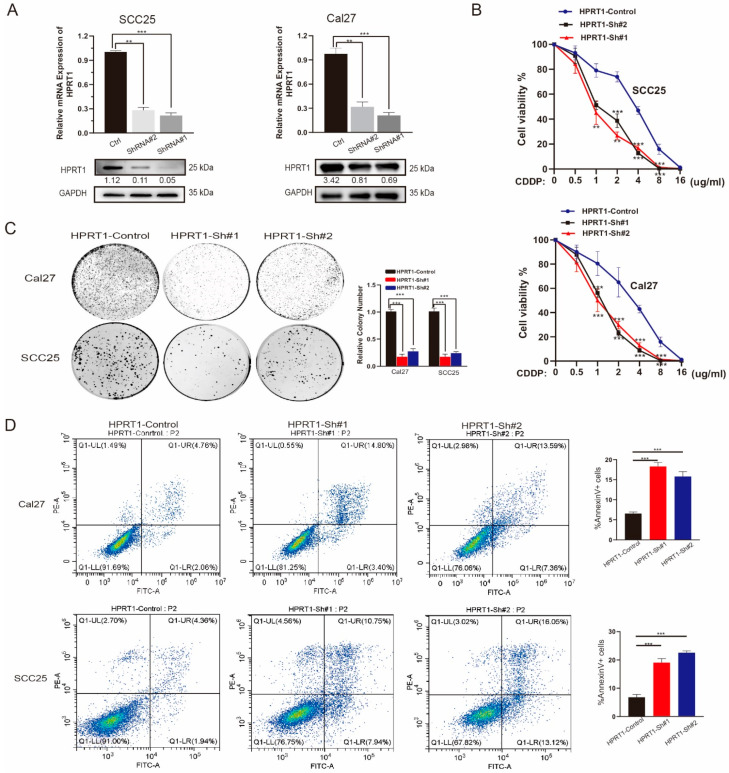
Knockdown of HPRT1 reduced cisplatin resistance of oral cancer cell in vitro. (**A**) The establishment of sh-HPRT1 stable cell lines was verified at the mRNA level and protein level in two OSCC cell lines (SCC25 and CAL27). (**B**) The chemoresistance of Cal27 (upper panel) and SCC25 cells (lower panel) with HPRT1 silencing or not were performed by cell viability assays under different gradients of doses of CDDP. (**C**) Colony formation assays of Cal27 and SCC25 cells with HPRT1 silencing or not under cisplatin treatment. (**D**) Flow cytometry of Annexin V-FITC/PI staining in SCC25 and Cal27 cells with HPRT1 silencing or not (left panel) and percentage of Annexin V+ cells (right panel). Values present the mean ± SEM. ** *p* < 0.01, *** *p* < 0.001. We repeated every experiments for 3 times.

**Figure 4 cancers-14-00855-f004:**
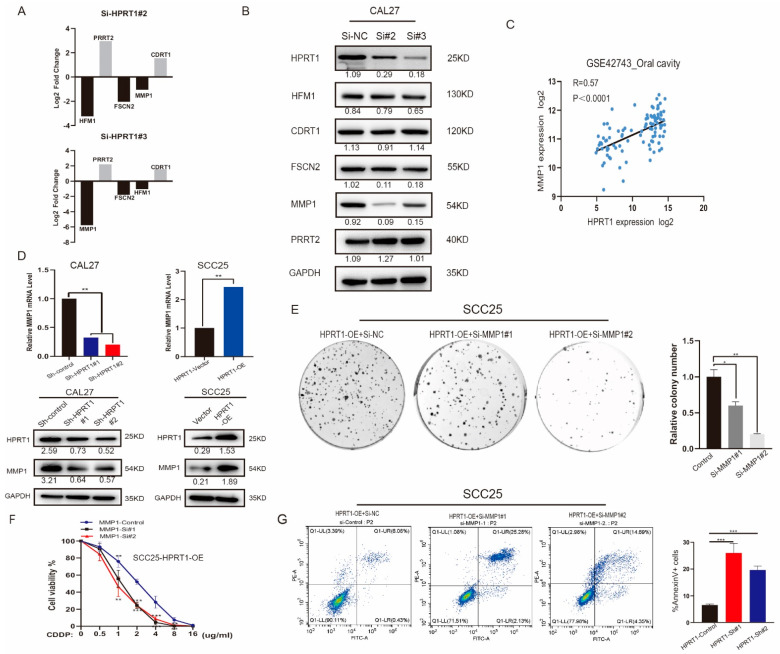
HPRT1 positively regulates the expression of MMP1 in oral cancer cell. (**A**) Top 5 differentially expressed protein coding gene genes (HFM1, CDRT1, TECRP1, FSCN2, MMP1) in RNA-Seq analysis (Cal27-si#2 and Cal27-si#3 versus Cal27#NC cells) were ranked by their log2 fold change. (**B**) Western blotting of showing protein expression of HFM1, CDRT1, TECRP1, FSCN2, MMP1 in cells with different expression levels of HPRT1. (**C**) Correlation analysis of mRNA expression levels of HPRT1 and MMP1 in GSE42743. (**D**) mRNA and protein expression levels of HPRT1 and MMP1 in Cal27 cells(Sh-control, Sh HPRT1#1, Sh-HRPT1#2) and SCC25 cells (HPRT1-Vector, HPRT1-OE). (**E**) Colony formation assays of SCC25-HPRT1-OE cells with MMP1 silence or not under cisplatin treatment. (**F**) The chemoresistance of SCC25-HPRT1-OE cells with MMP1 silencing or not were performed by cell viability assays under different doses of CDDP. (**G**) Flow cytometry of Annexin V-FITC/PI staining in SCC25-HPRT1-OE cells with or without MMP1 silencing (left panel) and proportion of Annexin V+ cells (right panel). Values present the mean ± SEM. * *p* < 0.05, ** *p* < 0.01, *** *p* < 0.001. We repeated every experiments for 3 times.

**Figure 5 cancers-14-00855-f005:**
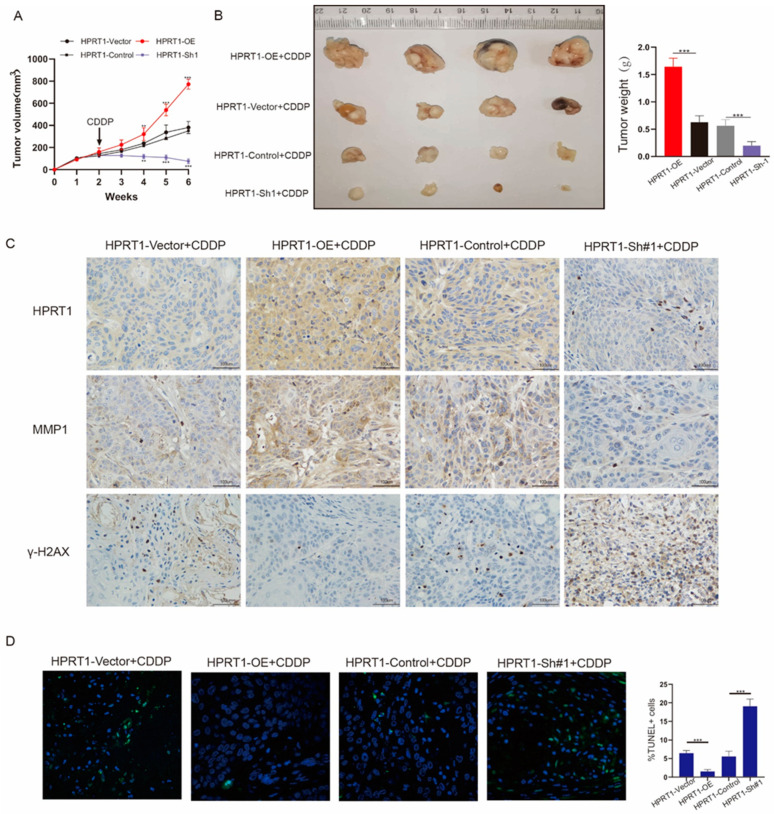
HPRT1 promote chemoresistance in vivo. (**A**) Cisplatin (4 mg/kg, twice a week) was injected to all groups of mice two weeks later after cells inoculation. The tumor growth volume was recorded once per week. (**B**) Tumors were gathered 6 weeks after inoculation. (**C**) The expression levels of HPRT1, MMP1 and γ-H2AX were determined by immunohistochemistry in the tumor tissues of each group of mice. (**D**) Representative images of TUNEL staining of tumors formed by the tumor tissues of mouse (left panel); proportion of TUNEL+ cells (right panel). Values present the mean ± SEM. ** *p* < 0.01, *** *p* < 0.001.

**Figure 6 cancers-14-00855-f006:**
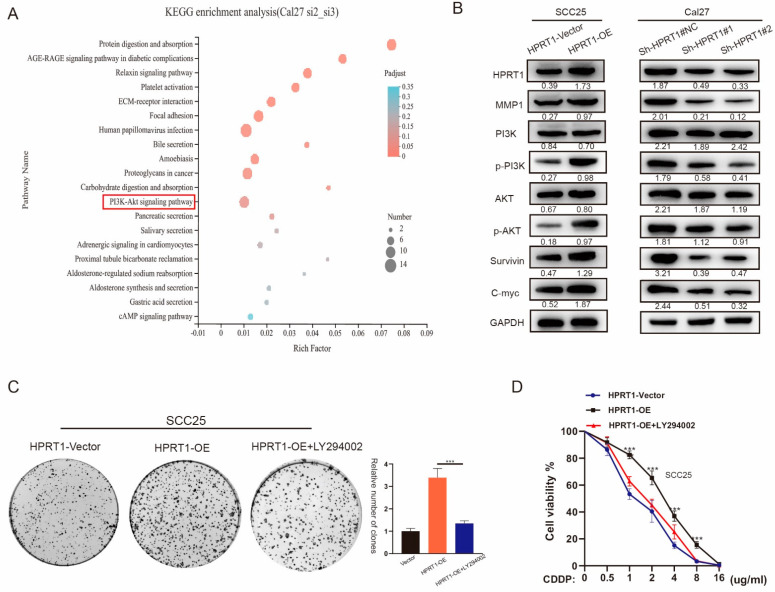
HPRT1 promotes CDDP chemoresistance of OSCC through activation of the PI3K/AKT pathway. (**A**) KEGG enrichment analysis based on RNA-Seq analysis (Cal27-si#2 and Cal27-si#3 versus Cal27#NC cells). (**B**) Proteins extracted from cells stably transfected with MMP1 were subjected to western blotting. The relative expression of p-AKT, Survivin and c-myc was examined. GAPDH served as the loading control. (**C**,**D**) Cells overexpressing HPRT1 were treated with LY294002, a PI3K/AKT inhibitor. Chemoresistance were determined by colony formation (**C**) and cell viability assays (**D**). Values present the mean ± SEM. *** *p* < 0.001.

**Figure 7 cancers-14-00855-f007:**
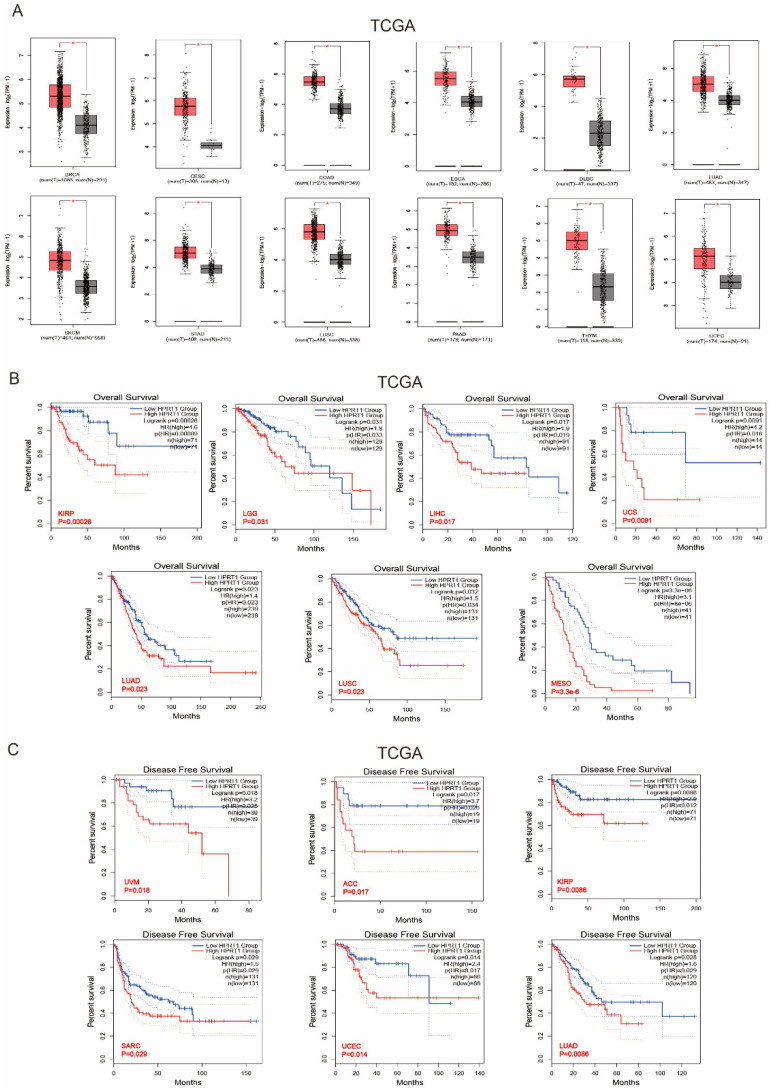
Clinical Value of HPRT1 in Other Cancer Types. (**A**) mRNA levels of HPRT1 in various cancer tissues compared with adjacent normal tissues in TCGA. (**B**) Kaplan-Meier overall survival analyses of patients expressing higher (red line) or lower (blue line) levels of HPRT1 RNA verified across pan-cancer types. (**C**) Kaplan-Meier disease free survival analyses of patients expressing different levels of HPRT1 RNA (red line represents high expression, and blue line represents low expression) verified across pan-cancer types. The cancer out-come-linked gene expression data were accessed and graphed on 2 June 2021. * *p* < 0.05.

## Data Availability

All datasets presented in this study are included in the Appendix A.

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
