# Peer review of "HPRT1 Promotes Chemoresistance in Oral Squamous Cell Carcinoma via Activating MMP1/PI3K/Akt Signaling Pathway"

_cancers, 2022, doi:10.3390/cancers14040855_

Round 1
Reviewer 1 Report
The authors investigate the role of HPRT in OSCC and more specific resistance to cisplatin in OSCC. They show that HPRT is associated with poor OS and DFS in HNSCC. In addition they claim that knockdown of HPRT is associated with better response to cisplatin which is associated with the PI3K/AKT pathway. Although overexpression of HPRT has been associated with poor prognosis in HNSCC (Ahmadi M et al.2021), but the link with cisplatin resistance was not shown. To proof the latter the author make use of combination of in vitro and in vivo techniques. The following points should be addressed:
The authors show that HPRT knockdown in SCC25 and CAL27 is equally efficient in both cell lines on mRNA level however on protein level the knockdown of CAL27 looks less efficient. Can the authors provide densitometry for every western blot?
In addition the difference in protein level does not reflect in lower sensitivity to CDDP(fig 3B,C). Can the authors explain this? Moreover, it is difficult to state that HPRT expression is a regulator of chemoresistance since it is not clear if the cell lines are resistant to CDDP. The authors should be cautious with making strong conclusions.
Moreover, Fig3D the gatings of the figures look different. What is the gating based on? It is also not clear what concentration of CDDP is used in fig 3C,D and how many repeats were performed? The authors should provide these detail for every experiment. It is also not clear what the baseline effect of HPRT knockdown is on these cell lines, since this is omitted in every panel of fig. 3. The authors should add this information the paper.
In Fig.4 the authors opt to send the siRNA treated CAL27 to RNA-seq, why did they not send the more stable shRNA cells to RNA-seq? In the same fig. (panel B and C) the authors claim a strong correlation between HPRT expression and MMP1, however the expression of HPRT does not correlate when we look between two siRNA conditions (see si2 vs Si3 levels of HPRT and MMP1).
Why do the authors opt for overexpression of HPRT in scc25, is this not already highly expressed as claimed in earlier in the paper? In addition HPRT-OE cells do not shown the same response to CDDP levels, the HPRT-OE cells looks more sensitive to CDDP then HPRT control cells (fig4F vs fig 3B). How can the authors explain this?
In fig 5 the authors shows the in vivo confirmation experiments using CAL27 cells. Can the authors show the in vitro data of the CAL27-OE vector? What does HPRT shRNA do the the growth of the tumors?
In Fig 6 the authors show a KEGG pathway mapping, how was the analysis done? The authors should add this the material and method section? In fig. 6B the authors again switch the shRNA for CAL27 and OE for SCC25, why did they opt for this? The authors should provide the information on both cell lines. In addition the sensitivity of SCC25 vector (fig6D) and SCC25 (fig3B) control is different, can the author elaborate on this? Moreover, the link between HPRT, MMP1 and PI3K is also not clear and not enough data is available to form strong conclusions.
Reviewer 2 Report
The authors present an interesting comprehensive study how HPRT1 promotes chemoresistance in oral squamous cell carcinoma. The mechanism of MMP1/PI3K/Akt Signaling Pathway is studied.
The topic is interesting as head and neck cancer incidence is growing.
The study includes both cell culture experiemnts as well as human carcinoma samples. The methodology is adequate and well described.
The resultes are nicely illustrated with numerous figures, graphs and tables. Immunohistochemical analysis: can you provide figures with higher magnification and clearly state the antigen localization on cellular level (nucler vs. cytoplasmatic). Some figures have too high background and so specificity is questionable.
The discussion is well written.
The refernces are up-to-date.
Round 2
Reviewer 1 Report
The authors answered the raised questions.